# Rapid increase in simulated North Atlantic dust deposition due to fast change of northwest African landscape during the Holocene

Sabine Egerer [1,2], Martin Claussen [1,3], and Christian Reick [1]

[1]Max Planck Institute for Meteorology, Bundesstraße 53, 20146 Hamburg, Germany
[2]International Max Planck Research School on Earth System Modelling, Bundesstraße 53, 20146 Hamburg, Germany
[3]Center for Earth System Research and Sustainability, Universität Hamburg, Bundesstraße 53, 20146 Hamburg, Germany

**Correspondence:** Sabine Egerer (Sabine.Egerer@mpimet.mpg.de)

**Abstract.**

Marine sediment records from a series of core sites along the northwest African margin show a sudden increase in North Atlantic dust deposition about 5 ka BP that has been associated with an abrupt end of the African Humid Period (AHP). To assess the causes of the abrupt shift in North Atlantic dust deposition, we explore changes in the Holocene dust cycle and in

North African climate and landscape by performing several time slice simulations from 8 ka BP until the pre-industrial era. To do this, we use the coupled aerosol-climate model ECHAM6-HAM2 including dynamic vegetation and interactive dust, whereas ocean conditions and lake surface area are prescribed for each time slice.

We find a rapid increase in simulated dust deposition between 6 and 4 ka BP that is fairly consistent with the abrupt change in marine sediment records at around 20° N close to the northwest African margin. At more northern and more remote cores, a

significant change in dust deposition is noticeable roughly between 6 and 2 ka BP in the simulations as well as in the records, but the change is less sharp compared to the near-margin core sites. The rapid change in simulated dust deposition is caused by a rapid increase in simulated dust emission in the western Sahara, where the main dust sources for dust transport towards the North Atlantic are located. The sudden increase in dust emission in the western Sahara is according to our simulations a consequence of a fast decline of vegetation cover from 22° N to 18° N that might occur due to vegetation-climate feedbacks

or due to the existence of a precipitation threshold on vegetation growth. Additionally, the prescribed gradual reduction of lake area enforces accelerated dust release as highly productive dust sources are uncovered. Concurrently with the continental drying, surface winds in the western Sahara are accelerated. Changes in the Saharan landscape and dust emission south of 18° N and in the eastern Sahara play a minor role in driving the dynamics of North Atlantic dust deposition at the core sites. Our study identifies spatial and temporal heterogeneity in the transition of the North African landscape. As a consequence,

implications from local data records on large scale climate have to be treated with caution.

## 1  Introduction

North Atlantic sediment records show an abrupt increase in dust accumulation close to the northwest African margin about 5 ka BP (deMenocal et al., 2000; Adkins et al., 2006; McGee et al., 2013a; Albani et al., 2015), which is also observed downwind in the tropical North Atlantic and at the Bahamas (Williams et al., 2016). The abrupt increase in dust deposition has been

attributed to an abrupt Holocene landscape change in North Africa. For instance, modeling studies reveal a sudden large scale decline of North African vegetation cover (Brovkin et al., 1998; Claussen et al., 1999; Liu et al., 2006). Also, paleohydrologic records point to a rapid drying-out of all water bodies at the end of the African Humid period (AHP) about 4.5 ka BP (Lézine et al., 2011). In contrast, pollen records from lake Yoa (Kröpelin et al., 2008), sediment records from the Manga Grasslands (Cockerton et al., 2014) and a modeling study of vegetation cover transition (Renssen et al., 2003) indicate a more gradual change of North African landscape. Paleohydrologic reconstructions show a southward retreat of the tropical rain belt with local differences in the timing and abruptness of the retreat (Shanahan et al., 2015). The origin of the abrupt change in North Atlantic dust deposition in the context of Holocene landscape change in North Africa is thus still a matter of debate.

In a previous simulation study (Egerer et al., 2017), we have tested the hypothesis that a gradual decline of North African vegetation and lake cover results in an abrupt increase in North Atlantic dust deposition either due to the nonlinearity in dust activation or due to the heterogeneous distribution of major dust sources. Our simulations revealed a gradual increase in North Atlantic dust deposition as a response to gradual landscape changes. This suggests that either a fast vegetation decline or a rapid desiccation of lakes is crucial to explain the abrupt increase in dust deposition in the marine sediment cores. Fast changes in vegetation cover have been attributed to a precipitation threshold on vegetation (Liu et al., 2006) or to feedbacks between climate and vegetation that amplify the gradual insolation forcing (Brovkin et al., 1998; Claussen et al., 1999). The first who proposed such a positive vegetation-climate feedback was Charney (1975). According to his theory, the high desert albedo leads to more stable conditions in the air column above compared to the surroundings and thus precipitation is suppressed, which results in a self-stabilization of the desert. Similarly, the mechanism works the opposite way in vegetated areas due to their low albedo. A change in external conditions may trigger an abrupt transition from a humid vegetated state to a hyperarid desert state in the presence of these positive feedbacks.

Previous simulation studies on the Holocene dust cycle (Sudarchikova et al., 2015; Albani et al., 2015; Egerer et al., 2016, 2017) did not include climate consistent calculations of the vegetation distribution ('dynamic vegetation') and were thus not suitable to analyze the link between changes in dust and vegetation during the Holocene. In this study, we explore the possible link between a sudden change of vegetation cover arising in interaction with the hydrological cycle and atmospheric dynamics and an abrupt change in North Atlantic dust deposition during the Holocene. The study is guided by the questions: Can we confirm an abrupt shift in North Atlantic dust deposition in our simulations as found in marine sediment records? How is the shift in North Atlantic dust deposition linked to Saharan landscape transition towards the end of the AHP? How does the timing and the abruptness of Saharan landscape and climate transition vary spatially? To answer these questions, we perform a series of time slice simulations from the mid-Holocene to the pre-industrial era, where vegetation, climate and dust are coupled dynamically in the global aerosol-climate model ECHAM6-HAM2 (version echam6.3-ham2.3). Ocean conditions for each time slice and a linear decline of lake surface area over time are prescribed.

Our study is structured as follows: First, we present a brief description of the model and experiment setup and evaluate the model against data. We then compare our simulated North Atlantic dust deposition to marine sediment records and demonstrate the link between changes in North Atlantic dust deposition and changes in Saharan dust emission, landscape and climate.

Further, we analyze the spatial heterogeneity of changes in dust emission and vegetation and analyze the importance of changes in atmospheric circulation. Finally, we discuss our findings and draw our main conclusions.

## 2 Model and experiment setup

### 2.1 Model description

We use the global aerosol-climate model ECHAM6-HAM2 (version echam6.3-ham2.3) (Stier et al., 2005; Stevens et al., 2013) at a model resolution of T63L31, which corresponds to a horizontal resolution of approximately 1.9°x1.9° and 31 vertical pressure levels in the atmosphere. The aerosols that are calculated interactively in the model are sulfate, black carbon, organic carbon, sea salt and mineral dust. In this study, we only focus on mineral dust. In the model, the impact of dust on the solar radiation budget through scattering and absorption and on the cloud structure is implement (for details see Stier et al. (2005)). Bare soils and areas covered with sparse vegetation, such as grasses, shrubs and crops are assumed to be potential dust sources in the model (Stanelle et al., 2014). Additionally, former paleolakes serve as preferential dust sources as fine grained material that is deposited in dried-out lake beds can be easily deflated by surface winds. If certain criteria are fulfilled (e.g. the soil has to be dry and uncovered), dust is emitted from these potential and preferential dust sources as soon as the surface wind velocity exceeds a critical threshold.

The dust emission scheme is based on Tegen et al. (2002). The main mechanism considered in this scheme is saltation bombardment. The size distribution of the emitted particles is prescribed via log-normal functions of a coarse (mass mean radius (mmr) = 1.75 $\mu$m, standard derivation $\sigma$ = 2 $\mu$m) and an accumulation mode (mmr = 0.37 $\mu$m, $\sigma$ = 1.59 $\mu$m). After exceeding a critical threshold, the horizontal dust flux is parametrized as a cubic function of the surface wind velocity following Marticorena and Bergametti (1995). A ratio between vertical and horizontal fluxes is given depending on particle size and surface properties. Dust is transported via tracers in the atmosphere component ECHAM. Deposition processes involve washing out of particles from the atmosphere by precipitation (wet deposition), turbulent downward mixing (dry deposition) and gravitational settling of particles (sedimentation).

In this model version, we use a 5-layer soil scheme to handle soil moisture. The charge of the skin water reservoir (thickness $10^{-6}$cm) is taken as a measure for the soil wetness. As long as the skin reservoir is filled, dust emission is suppressed. Also, lakes disable dust emission. If the lake surface fraction of a grid cell is above 50 %, the grid cell is handled as a lake cell and no dust emission from this grid cell is possible.

Vegetation is described as a composition of 11 plant functional types (PFTs) in the land surface model JSBACH of ECHAM including grass and woody (trees and shrubs) types. Natural land cover change and vegetation dynamics is simulated by the DYNVEG component (Brovkin et al., 2009; Reick et al., 2013). The simulation of vegetation in JSBACH is based on the 'universal presence principle', i.e. each PFT can potentially grow everywhere. Bioclimatic limits restrict the establishment of PFTs. Several processes control land cover change: PFT cover can be reduced by natural death or disturbances, e.g. through wildfires, thereby releasing space for migration of other vegetation. PFT cover can increase through migration of plant species in the released space, the so called 'uncolonized land'. In general, vegetation establishment is possible, when net primary

productivity (NPP) is positive at least for some years. Different PFTs compete for the uncolonized land and their success depends on growth form and productivity, where more productive plants have a competitive advantage. At first, grass PFTs have an advantage because they grow much faster than woody types (trees and shrubs). In absence of disturbances, woody PFTs are in favor of grass PFTs due to light competition. Finally, the fraction of a grid cell inhospitable to vegetation ('bare land') may expand or shrink depending on climate conditions as measured by growth success.

For this study, a simple dynamic soil albedo scheme has been included in the model, where the soil albedo $\alpha_{soil}$ is parametrized based on plant net primary productivity (NPP):

$$\alpha_{soil} = \alpha_{obs} - (\alpha_{obs} - \alpha_{dark} \cdot \min\left(\frac{\overline{NPP}}{NPP_{dark}}, 1\right)), \tag{1}$$

where $\alpha_{obs}$ is the standard soil albedo in JSBACH, which is based on present-day observations, while $\alpha_{dark}$ is the albedo of soil by the presence of soil organic carbon as measured by the average normalized net primary productivity $NPP_{dark}$, when vegetation is composed half of tropical summer-green trees and half of rain-green shrubs. Eq. (1) is evaluated separately for the visible and near infra-red range. $\alpha_{dark}$ is set to 0.13 in the visible range and to 0.22 in the near infra-red range. $\overline{NPP}$ is the 5-years average of the actual $NPP$ (Zink, 2014).

## 2.2 Experiment setup

We perform a series of time slice simulations covering the Holocene from 8 ka BP to the pre-industrial era, 1850 AD, in 2 ka intervals, where, for example, 6k refers to 6 ka BP. The mid-Holocene time slice is hereafter referred to as 6k, the pre-industrial time slice as 0k. We chose the pre-industrial era rather than present-day to exclude any anthropogenic impact. Because marine sediment records indicate a rather abrupt change in dust accumulation at about 5.5 and 4.9 ka BP (deMenocal et al., 2000; McGee et al., 2013a), we add a time slice at 5 ka BP. Vegetation and dust are calculated interactively.

Paleohydrologic records show that the maximum of deep lake formation occurred during the early Holocene around 9 ka BP, whereas the maximum extent of water surface was reached only at 6 ka BP and fell thereafter (Lézine et al., 2011). Thus, we prescribe the lake surface area identically for the 8k and 6k time slice in our simulations based on a paleolake reconstruction (Tegen et al., 2002) (Fig. 1). Thereafter, the lake surface area is prescribed to decline linearly in North Africa similar to the first scenario in Egerer et al. (2017). Pre-industrial lake surface area is prescribed based on satellite data (Loveland et al., 2000) (Fig. 1). Orbital forcing parameters for each time slice are set following Berger (1978). Changes in greenhouse gas concentrations are assumed to have a minor impact and are set constant across all time slices following the PMIP protocol for the mid-Holocene (Harrison et al., 2001). We prescribe sea surface temperatures (SST) and sea ice concentration (SIC) for each time slice from 50 years averages calculated interactively in a transient Holocene simulation with MPI-ESM1 that does not contain interactive dust (Bader et al., in prep.). This simulation includes orbital and $CO_2$ forcing and captures the Holocene era from 8 ka BP until present. We use 50-years mean vegetation cover and cover fractions of all 11 PFTs from the model output of this transient Holocene simulation to initialize the vegetation distribution in North Africa (17° W–40° E; 10° N–30° N) for each time slice in our study. To bring the vegetation distribution in equilibrium, we first run 30 years with an accelerated vegetation dynamics followed by 570 years with vegetation dynamics at normal speed. We take the last 200 years for evaluation. We consider a

rather long period for evaluation because the dynamic vegetation in JSBACH varies on time-scales of more than 100 years as for example in the grass and shrub fraction.

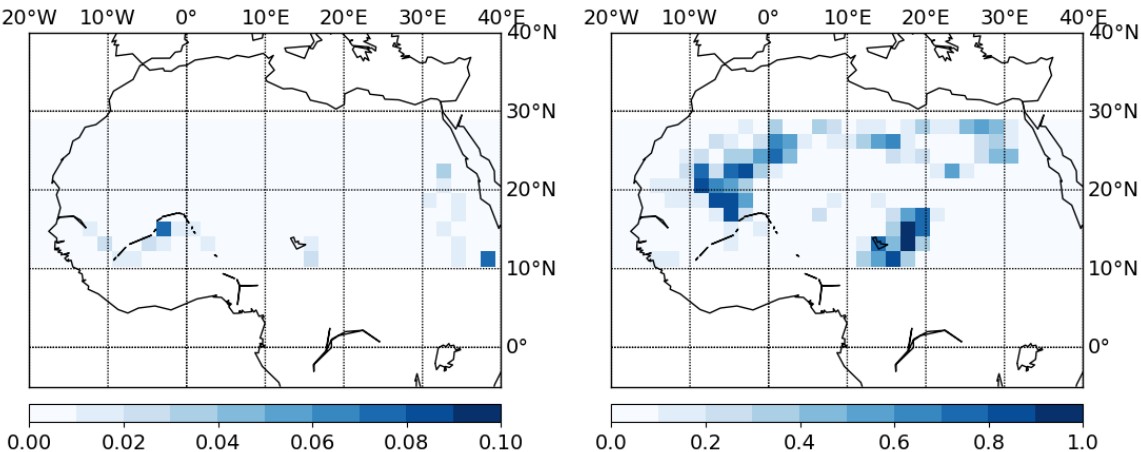

**Figure 1.** Pre-industrial lake distribution based on satellite data (Hagemann, 2002) (left) and a paleolake reconstruction of Tegen et al. (2002) (right). The color indicates the lake fraction in a grid cell. Note that the scales are different.

## 2.3 Model evaluation

### 2.3.1 Vegetation cover

We compare the vegetation distribution that was simulated dynamically in our model to a biome reconstruction based on pollen records of the BIOME 6000 data set (Harrison, 2017). To do so, we translate the vegetation cover fractions and the PFT cover fractions from our 0k and 6k simulations into biomes applying the method of Dallmeyer et al. (2018) (Fig. 2).

In the pre-industrial (0k) simulation, tropical forest reaches up to 12° N in agreement with remote sensing data (Ramankutty and Foley, 1999). Between 12° N and 14° N, the simulation shows a mixture of savanna, grassland and desert vegetation. The

northward extent of our simulated vegetation seems a bit underestimated compared to satellite data entering the land surface data set of Hagemann (2002). Also pollen records close to the coast indicate savanna vegetation up to 16° N and grassland even north of 20° N (Fig. 2).

In the mid-Holocene (6k) simulation, the surface is densely covered with tropical forest up to 14° N and with savanna up to 21° N in the western Sahara and up to 16° N in the eastern Sahara. Grassland reaches up to 23° N in the western Sahara

and up to 19° N in the eastern Sahara. Pollen records predict grassland even up to 27° N. The northern vegetation extent is underestimated in our mid-Holocene simulations compared to data. In contrast, between 14° N and 17° N, pollen records predict grassland, whereas in our simulations, savanna vegetation is dominant.

While the simulated pre-industrial vegetation distribution turns out to be shifted southward by about 2° compared to satellite data, the mid-Holocene northward extent of vegetation is clearly underestimated, only reaching up to between 19° N and 23° N compared to 27° N obtained from reconstructions. Nevertheless, overall, the mid-Holocene vegetation is shifted significantly northward by about 6° compared to the pre-industrial vegetation in our simulations.

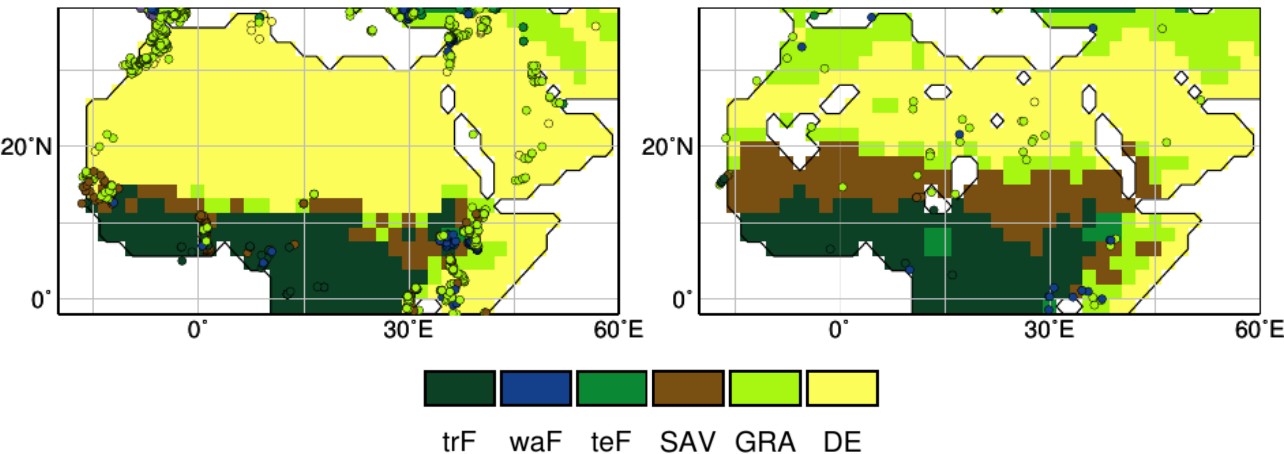

**Figure 2.** Biomes determined from simulated vegetation cover fractions and PFT cover fractions applying the method of Dallmeyer et al. (in prep.) for 0k (left) and 6k (right). The circles indicate biomes reconstructed from pollen data of the BIOME 6000 data base (Harrison, 2017). The white areas within the continent are lakes. (trF = tropical forest, waF = warm-mixed forest, teF = temperate forest, SAV = savanna, GRA = grassland, DE = desert)

### 2.3.2 West African monsoon

It is known that ECHAM6 has a dry bias in the Sahel region during the summer for present-day climate and the West African Monsoon (WAM) does not extend sufficiently far north (Eichhorn and Bader, 2016) which is consistent with the somewhat underestimated northward extent of vegetation in our pre-industrial simulation (Fig. 2).

The simulated precipitation changes between mid-Holocene and pre-industrial times obtained in our study clearly exceed those of the precipitation records from proxy data between 10° N and 15° N by about 800mm/yr and precipitation changes in other modeling studies by about 1000mm/yr (Perez-Sanz et al., 2014) (Fig. 3). Only one study with prescribed vegetation and a prescribed reduction of dust AOD predicts higher precipitation changes compared to our study (Pausata et al., 2016). Between 15° N and 20° N, the simulated precipitation changes are slightly lower than those of observations and Pausata et al. (2016) and slightly higher than predicted by all other CMIP5 models. Still, they lie within the uncertainty range of the observations. North of 20° N, the simulated precipitation changes are far below the reconstructed precipitation changes. The lack of vegetation north of 20° N is consistent with an underestimation of the simulated northward extent of the West African monsoon during

the mid-Holocene compared to paleorecords (Bartlein, 2011; Perez-Sanz et al., 2014) (Fig. 3). Rather than an underestimation of absolute mid-Holocene precipitation changes as in the CMIP5 models, we find a mismatch in the meridional gradient of precipitation changes between our simulations and the reconstructions from proxy data (Bartlein, 2011).

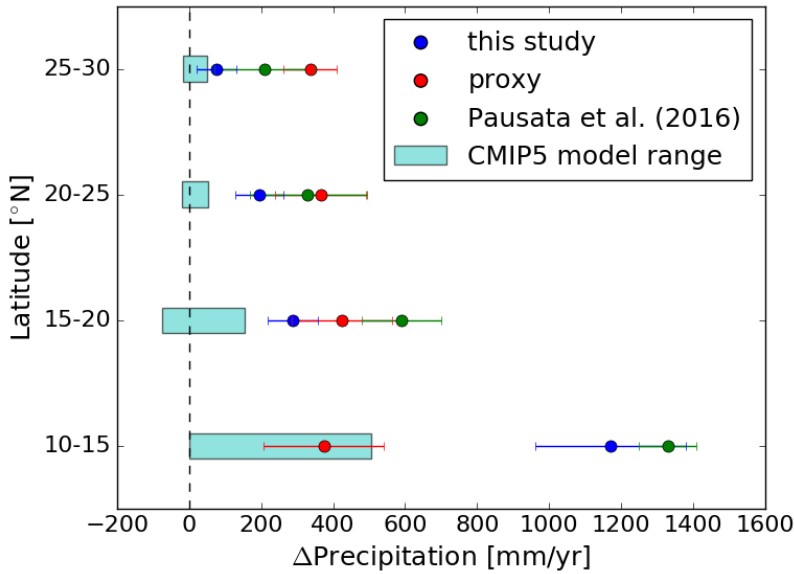

**Figure 3.** Comparison of reconstructed and simulated precipitation changes between the mid-Holocene and the pre-industrial era. The simulated zonal mean (17° W–40° E) anomalies between mid-Holocene and pre-industrial precipitation from this study are averaged over 5° latitudinal bands between 10° N and 30° N at the location of grid cells where reconstructions are available. Proxy data are from a reconstruction of precipitation from Bartlein (2011). Further, the CMIP5 model range (Harrison et al., 2015) and model values from Pausata et al. (2016) are averaged for the grid cells where reconstructions are available. Temporal standard deviations are given for our study and for Pausata et al. (2016) and spatial variation is indicated for the proxy data.

## 3 Results

### 3.1 Comparison with marine sediment records

We verify our simulated dust deposition by comparing model results with marine sediment records close to the northwest African margin and more remote in the North Atlantic. The locations of the considered core sites are given in Fig. 4 and Table 1. At the marine sediment cores, the dust flux was calculated as the difference between the total flux and the carbonate, opal and organic carbon flux. In Adkins et al. (2006), McGee et al. (2013a) and Williams et al. (2016), the $^{230}$Th normalization method was used to determine dust fluxes. Additionally, the studies by McGee et al. (2013a) and Williams et al. (2016) use endmember modelling to separate eolian and hemipelagic fluxes. Thereby, the coarse endmembers (approxiamately between 8 $\mu$m and 80 $\mu$m in size) are assumed to characterize eolian dust. Albani et al. (2015) provide an updated observational dataset of the sediment records of McGee et al. (2013a) with higher temporal resolution and information about particle size distribution. Here, we compare our simulation results to the total eolian fluxes and not to the <10 $\mu$m which are additionally presented by Albani et al. (2015). The dust data of Middleton et al. (2018) are constrained using the $^{3}$He$_{ET}$ flux proxy.

The marine records show an abrupt shift in dust deposition at the locations of the southern cores ODP658 and GC68 close to the northwest African margin, where the most prominent increase in dust deposition occurs at around 5.5 to 4.9 ka BP (deMenocal et al., 2000; Adkins et al., 2006; McGee et al., 2013a; Albani et al., 2015) (Fig. 5). At the locations of the more northern cores GC37 and GC49 the change in dust deposition is rather moderate. In our simulations, we find a rapid increase in North Atlantic dust deposition synchronous for several core sites along the northwest African margin (Fig. 5). At the southern cores, the quantitative range of simulated dust deposition agrees fairly well with the marine records, but the change in simulated dust deposition is less abrupt. The steepest rise in simulated dust deposition occurs between 5 and 4 ka BP. Additionally, a strong change occurs between 6 and 5 ka BP and, at least at the grid cells close to GC68 and ODP658, there is a moderate change between 4 and 2 ka BP. After 2 ka BP, dust deposition remains approximately constant until pre-industrial times. At the northern cores, the simulated dust deposition is much higher compared to the marine records for all times slices and we find a significant and rapid change between 6 and 4 ka BP in the simulations. At the locations of the the remote sediment cores VM20-234 in the tropical North Atlantic (TNA) and 103GGC at the Bahamas (Williams et al., 2016) the simulated dust deposition increases gradually between 6 and 2 ka BP. From 2 ka BP until pre-industrial, depositions increase further in VM20-234 and remain constant in 103GGC. For both, the relative change in dust flux is in fair agreement with marine dust records (Fig. 4). Absolute mass fluxes during the mid-Holocene are also reproduced in our simulations, whereas simulated pre-industrial dust fluxes are higher by factor two for 103GGC and by factor three for VM20-234 compared to the marine records. At GGC6 in the mid-North Atlantic (Middleton et al., 2018) we find a somewhat steeper increase of simulated dust deposition between 6 and 4 ka BP, whereas the records show a rise between 4 and 2 ka BP. Also, the absolute amount of simulated dust flux is higher by an order of magnitude. At all core sites, the simulated dust deposition fluxes remain nearly constant between 8 ka BP and 6 ka BP, which is consistent with the sediment records.

The simulated deposition flux at the core sites along the northwest African margin increases from north to south by factors between 2.5 and 3.1 from about 4–5.5 g/m²/a at 8 ka BP to 10–18 g/m²/a at 0 ka BP (Fig. 5). These factors are in agreement with those indicated by sediment records and also with those determined in a previous study (Egerer et al., 2016).

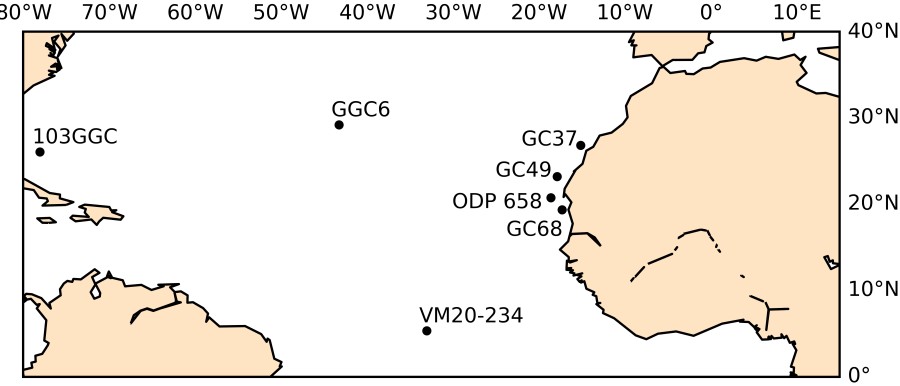

**Figure 4.** Site locations of marine sediment cores in the North Atlantic.

| Site | lat [°N] | lon [°W] | Reference |
|------|----------|----------|-----------|
| GC 37 | 26.82 | 15.12 | McGee et al. (2013a); Albani et al. (2015) |
| GC 49 | 23.21 | 17.85 | McGee et al. (2013a); Albani et al. (2015) |
| ODP 658 | 20.75 | 18.58 | Adkins et al. (2006) |
| GC 68 | 19.36 | 17.28 | McGee et al. (2013a); Albani et al. (2015) |
| VM20-234 | 5.33 | 33.03 | Williams et al. (2016) |
| 103GGC | 26.06 | 78.03 | Williams et al. (2016) |
| GGC6 | 29.21 | 43.23 | Middleton et al. (2018) |

**Table 1.** Site locations and data references of marine sediment cores in the North Atlantic.

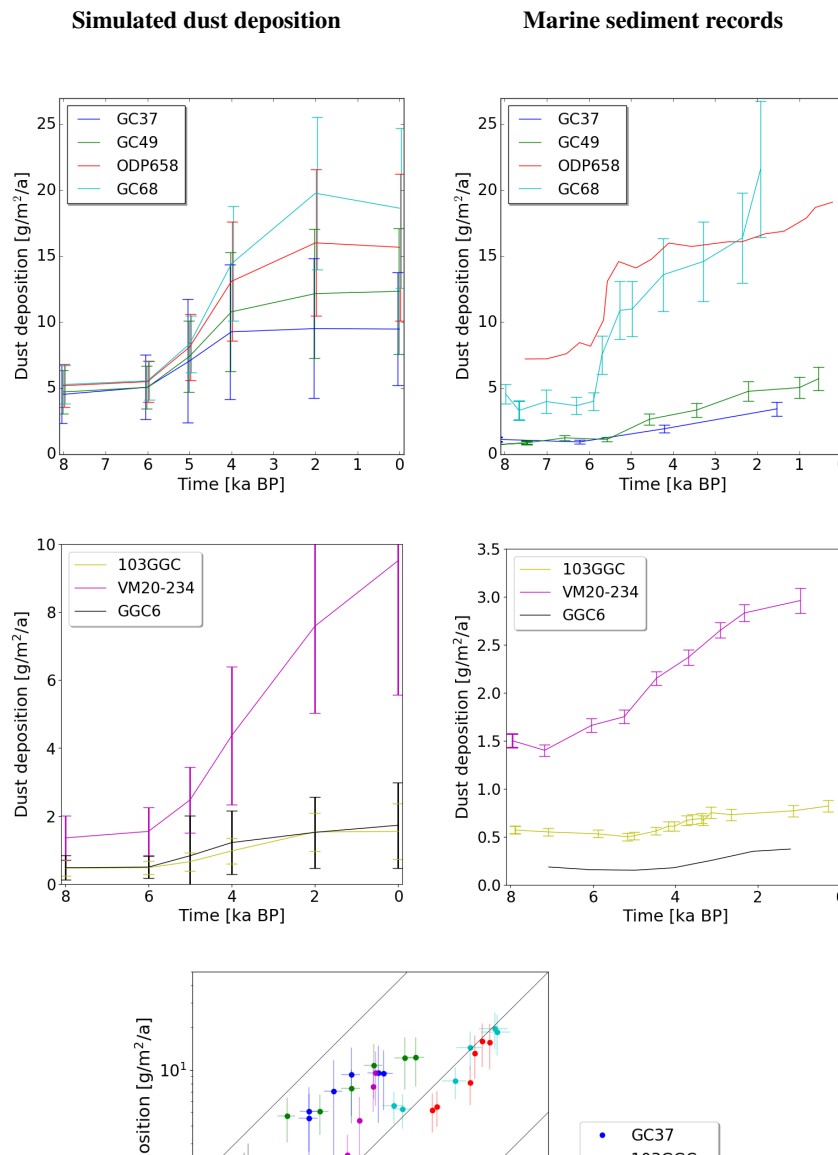

**Figure 5.** Comparison between simulated and observed dust fluxes at the position of the considered marine sediment cores. Left column: simulations. Right column: observations. Note that the scale is different for the remote core sites (plots in second row). For the data references see Table 1.

## 3.2 Link between North Atlantic dust deposition and Saharan dust emission

Satellite observations show that today the main dust plume is located north of 15° N (Prospero et al., 2002; Engelstaedter et al., 2006) originating from major western Saharan dust sources in Mauritania, Mali and southern Algeria as well as from the Bodélé Depression (Middleton and Goudie, 2001; Prospero et al., 2002; Engelstaedter et al., 2006). These regions are clearly visible in our simulated pre-industrial dust emission patterns (Fig. 6). Similarly, changes in dust emission between the mid-Holocene (6k) and pre-industrial (0k) simulation are maximal in the regions of today's major dust sources: the western Sahara and the Bodélé Depression (Fig. 6). In these regions, paleolakes existed during the mid-Holocene and inhibited dust emission. As soon as the lakes dried out, fine grained material was exposed to surface winds, which has made these areas very productive dust sources.

We aim to better understand the spatial relation between North African dust sources and North Atlantic dust deposition. The maximal changes in North Atlantic dust deposition occur between 6 and 4 ka BP in our simulations (Fig. 5). Hence, we show the changes in Saharan dust emission within this time period (Fig. 6). Whereas major changes in dust emission occur as far south as 12° N between 6 and 0 ka BP, the areas where changes in dust emission occur between 6 and 4 ka BP are almost exclusively located north of 18° N. Only in specific grid cells in the area of the Bodélé Depression and close to the Red Sea we find larger changes in dust emission before 4 ka BP. However, we argue that these dust source areas do not affect dust deposition at the marine cores close to the northwest African margin for the following reasons: In our simulations, we find that dust emission from the area around the Bodélé Depression (14° E–22° E; 12° N–18° N) does not significantly increase between 6 and 4 ka BP, but only after 4 ka BP (Fig. 7). In contrast, dust emission from the western Sahara rises rapidly between 6 and 4 ka BP in line with the rapid increase in dust deposition at the marine cores (Fig. 5). Observational evidence also supports our argument: Northeasterly surface winds mainly transport dust from western Saharan dust sources to the Atlantic (Cockerton et al., 2014). Further, the transport pathways from the Bodélé Depression, that were determined by a Lagrangian advection model (Washington et al., 2009; Ben-Ami et al., 2010), are located too far south to affect the marine cores.

We conclude from these findings and from Fig. 6 and Fig. 7 that the rapid change in simulated dust deposition between 6 and 4 ka BP at the position of the marine cores is directly linked to changes in western Saharan dust emission approximately between 15° W and 2° E and north of 18° N. In the following, we calculate the evolution of Saharan dust emission from this specific area (15° W–22° E; 18° N–29° N, marked with a grey frame in Fig. 6) between 8 and 0 ka BP and we investigate corresponding changes in Saharan vegetation, lakes and climate in the area to better understand the causes for the rapid change in North Atlantic dust deposition.

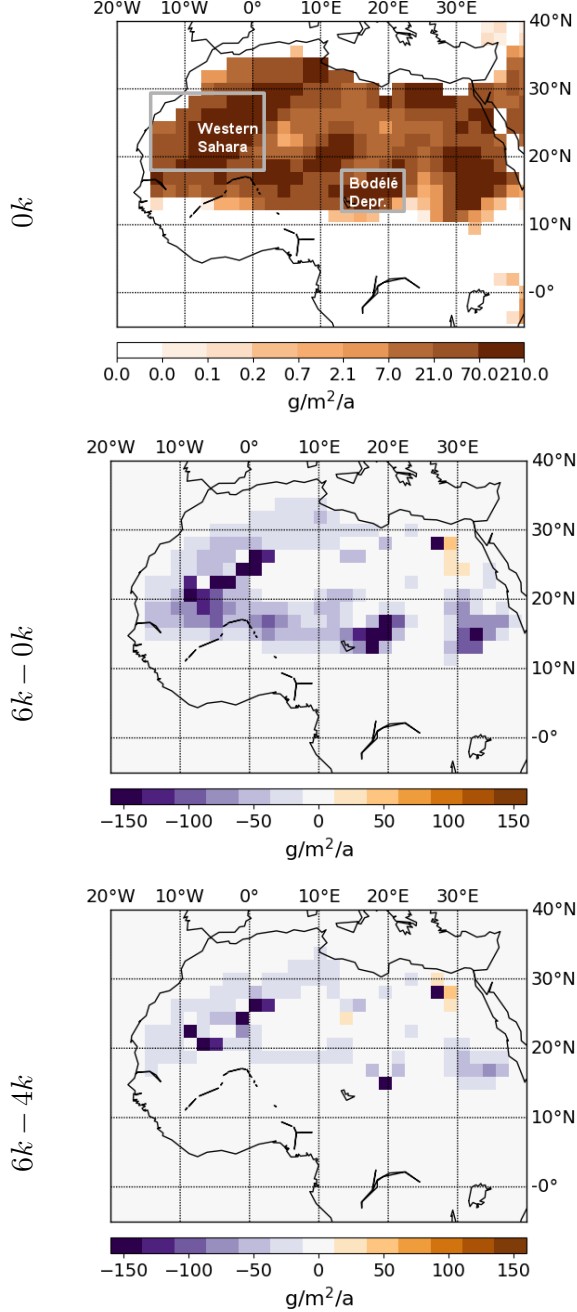

**Figure 6.** Simulated dust emission flux for $0k$ and changes in simulated dust emission flux between selected time slices ($6k - 0k$, $6k - 4k$). The areas of the western Sahara and the Bodélé Depression include the major dust sources today and are marked with a grey frame.

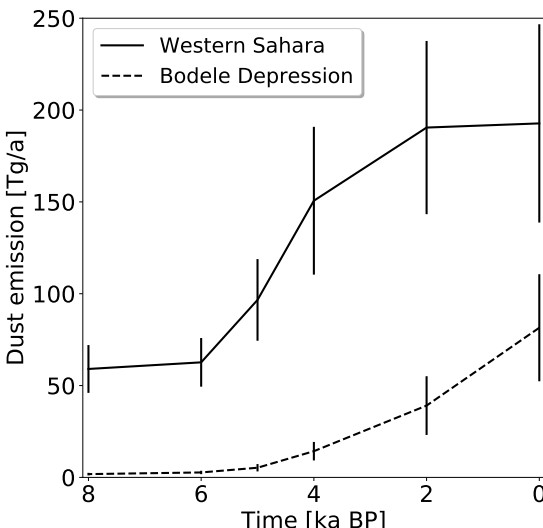

**Figure 7.** Simulated dust emission from the western Sahara (15° W–2° E; 18° N–29° N) (solid) and from an area including the Bodélé Depression (14° E–22° E; 12° N–18° N) (dashed). Both areas are shown in Fig. 6.

### 3.3 Causes for the rapid change in North Atlantic dust deposition

Considering the area of major changes in dust sources identified in the previous section (15° W–2° E; 18° N–29° N), simulated dust emission from the western Sahara remains constant between 8 and 6 ka BP, rises strongest between 6 and 4 ka BP, thereafter increases less strongly until it remains constant between 2 and 0 ka BP (Fig. 8). The evolution of simulated dust emission in the western Sahara during the Holocene is similar to the one of simulated dust deposition at the grid cells around the southern cores GC68 and ODP658 with a strong and rapid change between 6 and 4 ka BP (Fig. 5). This points to a strong coupling of western Saharan dust sources and dust deposition at the location of the southern dust cores at around 20° N. Northeasterly surface winds transport large amounts of dust from western Saharan sources to the North Atlantic.

In this western Saharan region, the vegetation cover fraction decreases monotonously between 8 and 2 ka BP, with the strongest reduction rate between 6 and 4 ka BP, before it remains approximately constant between 2 and 0 ka BP (Fig. 8). There is a clear link between the most prominent decline in vegetation cover from about 0.24 at 6 ka BP to about 0.1 at 4 ka BP and the rapid increase of western Saharan dust emission and associated North Atlantic dust deposition during this time. As vegetation cover decreases, larger areas are available for dust emission. Precipitation decreases in a similar manner as the vegetation (Fig. 8), with the strongest reduction in precipitation between 6 and 4 ka BP. Also, prescribed lake levels decrease quite strongly: During the mid-Holocene lakes covered about 25% of the surface, whereas today, there are barely any lakes in the western Sahara.

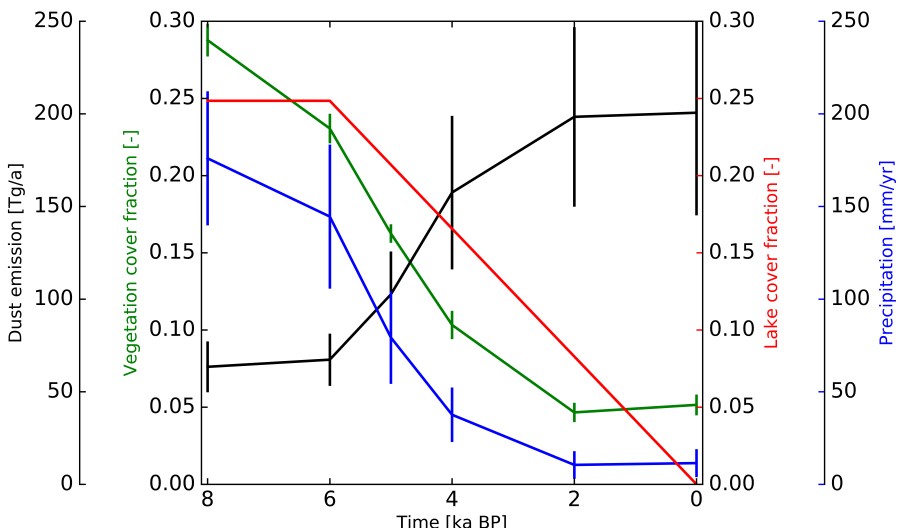

**Figure 8.** Simulated dust emission (black), vegetation cover fraction (green), lake cover fraction (red) and precipitation (blue) averaged over the western Sahara (15° W–2° E; 18° N–29° N).

To get a better understanding of the spatial coupling between changes in dust emission, vegetation and lake evolution and changes in climate, we divide North Africa (17° W–10° E; 10° N–30° N) in boxes of 2x2 grid cells, roughly 400x400 km (Fig. 9). This resolution is sufficient to identify spatial differences and yet all details are visible in the map. For each box, we plot dust emission, vegetation cover fraction, precipitation and lake cover fraction with similar axes as in Fig. 8.

South of 18° N (row D and E), dust emission does not increase significantly until 4 ka BP in the western Sahara, which justifies the choice of the area for evaluation (Fig. 6). The steepest rise in dust emission between 6 and 4 ka BP occurs in boxes A5, B2, B4 and C3. This region includes dried-out paleolake basins with fine grained sediments, which can be easily deflated by near surface winds. Accordingly, we prescribed a strong but gradual reduction in lake surface area in boxes A5, B2-B5, C2-4 and D3 (marked in blue/purple in Fig. 9). A strong shift in vegetation cover is restricted to boxes in row C,D and E. North

of them, vegetation cover remains low throughout the Holocene. In row C, we find a fast decline of vegetation cover between 6 and 4 ka BP in the western Sahara, which is clearly linked to a sudden rise in dust emission and associated dust deposition at the southern cores GC68 and ODP 658 at around 20° N (marked in purple in Fig. 9). In row D, the decrease of vegetation cover is even stronger and occurs later, from around 5 to 2 ka BP. Here, the vegetation density is too high to enable dust emission until 4 ka BP. In row E, the vegetation fraction remains high until present and inhibits dust emission nearly completely before

2 ka BP. The rapid decline of vegetation cover in row C and D is in line with the rapid decrease of precipitation. Also, in boxes B2-B4, the reduction of precipitation occurs fast between 6 and 4 ka BP and results partly from the strong lake surface reduction and partly from the moderate vegetation decline.

In the eastern Sahara, dust emission remains nearly constant or increases only slightly in most of the grid cells north of 18° N. In the very east, close to the Arabian Peninsula, there is a stronger increase in dust emission in boxes C13, D13 and D14 coinciding with a strong decline of vegetation and precipitation in row D (marked in green in Fig. 9). Here, the rise in dust emission occurs too late and the area is too remote to affect dust deposition at the core sites. In boxes D9 and D10, dust

5  emission rises gradually from 5 ka BP onwards to extremely high values in line with a strong fall of lake area, vegetation and precipitation (marked in purple in Fig. 9). This area contains the Bodélé Depression, the world's largest dust source today (Prospero et al., 2002). As discussed in section 3.2, dust is transported from the Bodélé Depression in direction of the West African Sahel and supply to the marine cores north of 19° N is minor (Washington et al., 2009; Ben-Ami et al., 2010; Cockerton et al., 2014). The high water levels during the mid-Holocene in boxes D9, D10, E8 an E9 point to the location of former lake

10  Megachad (Armitage et al., 2015). The strong reduction in lake surface area in this region does not result in a strong increase in dust emission before 4 ka BP (Fig. 7), because the vegetation cover is still too dense.

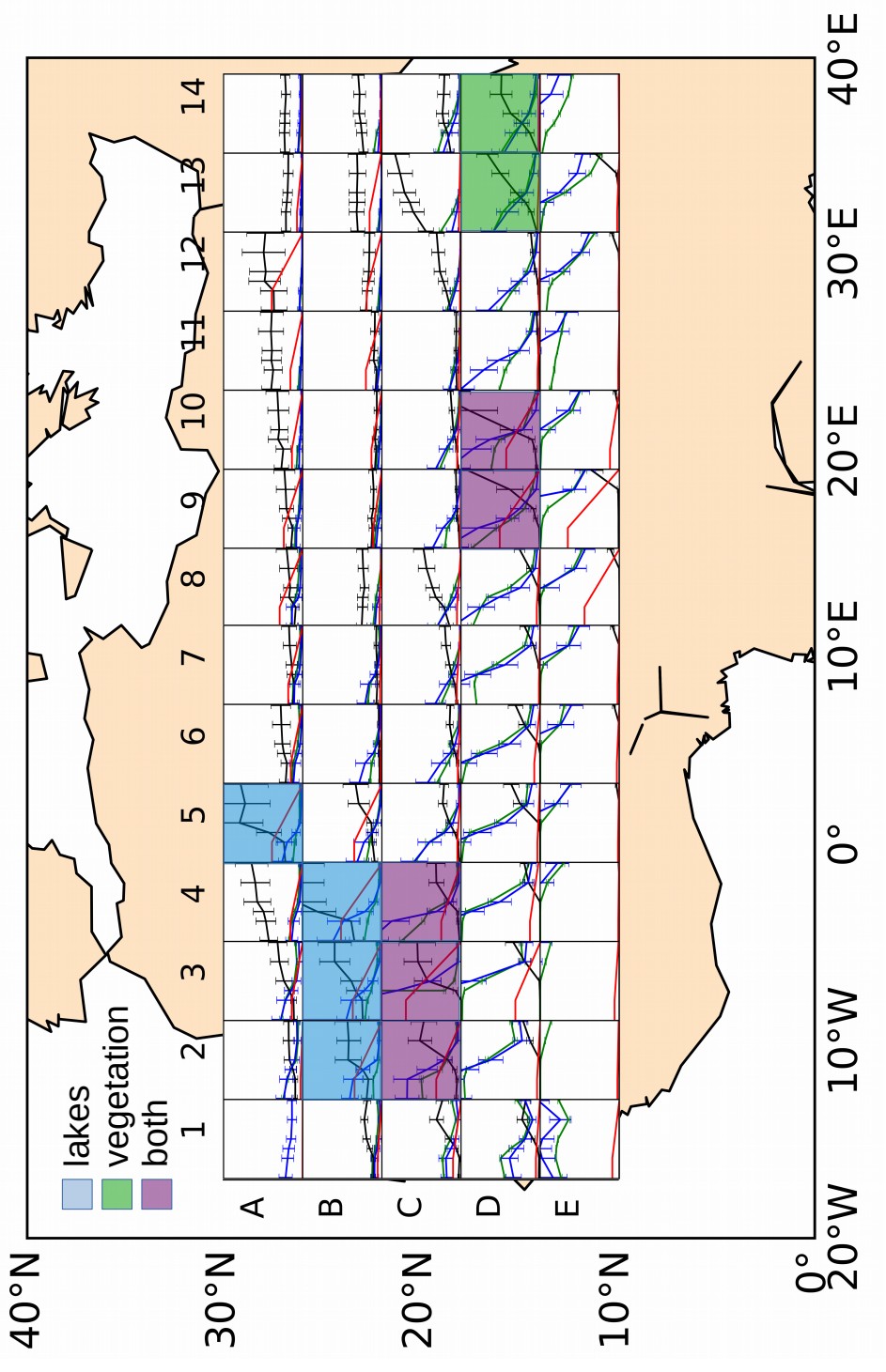

**Figure 9.** Dust emission (black), vegetation cover fraction (green), lake cover fraction (red) and precipitation (blue) in boxes of 2x2 grid cells in North Africa (17° W–39° E; 10° N–30° N). The scales are similar to those in Fig. 8. The boxes, where strong changes in dust emission coincide with strong changes in lake level (blue), vegetation cover (green) or both (purple) are marked.

### 3.4 Changes in the seasonal cycle of dust emission and atmospheric circulation

Here, we aim to get a better understanding of changes in the seasonal cycle of dust emission that is closely linked to changes in wind speed and direction. We find in all simulations that dust emission in the western Sahara is maximal during early spring (FMA) (Fig. 10). We refer hereby to the area that was identified in section 3.2 as dominantly impacting the changes in North

Atlantic dust deposition at the considered core sites. During this season, northeasterly trade winds between 12° N and 30° N transport dust from the western Sahara in altitudes up to 1000m height to the North Atlantic (Fig. 11) (Engelstaedter and Washington, 2007). At the same time, the maximal reduction in western Saharan dust emission occurs in early spring (Fig. 10) consistent with major changes in dust source areas (see section 3.2). In the mid-Holocene simulation, northeasterly winds at 925 hPa are reduced by up to 2 m/s along the northwest African margin and the western Sahara compared to the pre-industrial

simulation (Fig. 11). We find that, similar to the rapid increase in dust emission in the western Sahara (Fig. 8) and associated dust deposition along the northwest African margin (Fig. 5), there is also a sharp increase in early spring wind strength in the western Sahara mainly between 5 and 4 ka BP (Fig. 12). After 4 ka BP, the wind strength remains on the same level and even decreases towards the present.

    During pre-industrial summer (JAS), the tropical rain belt is shifted northward (Patricola and Cook, 2007; Gaetani et al.,

2017). Southwesterly winds transport moist air from the Equatorial Atlantic to the continent in accordance with a northward propagation of the West African monsoon. Dust is transported westward to the ocean within the Saharan Air Layer (SAL) which is connected to the African Easterly Jet (AEJ) in higher altitudes up to 500 hPa (Prospero and Carlson, 1972). Dust fallout from the SAL is incorporated into the northeasterly trade winds and transported westward along the coast, which is clearly visible in our simulations (Fig. 11). Consistently, we find a second smaller peak in western Saharan dust emission

during the summer (JAS) in our simulations (Fig. 10). During mid-Holocene summer, precipitation extended further north as shown in section 2.3.2 indicating a northward displacement of the West African monsoon and the Hadley circulation as described in previous studies (Merlis et al., 2013; D'Agostino et al., 2017; Gaetani et al., 2017). Consistently, winds at 925 hPa are reduced by up to 2 m/s south of 18° N in our simulations. Changes in simulated northeasterly winds occur mainly between 6 and 4 ka BP during the summer (Fig. 12). Changes in simulated dust emission in the western Sahara during the summer are

minor compared to the changes in early spring (Fig. 10).

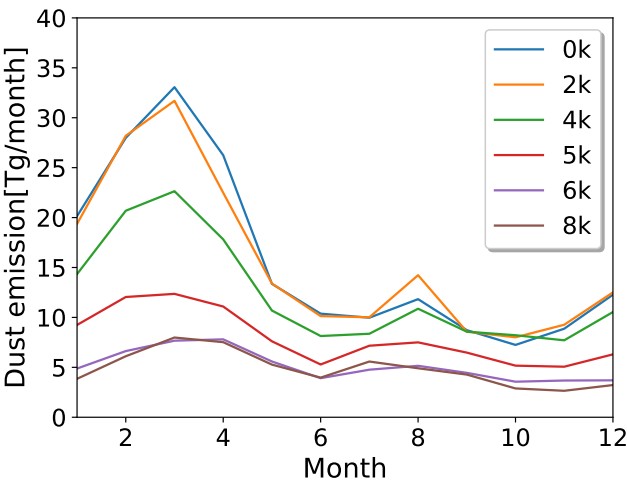

**Figure 10.** Mean seasonal cycle of simulated dust emission in the western Sahara (15° W–2° E; 18° N–29° N) for all time slices.

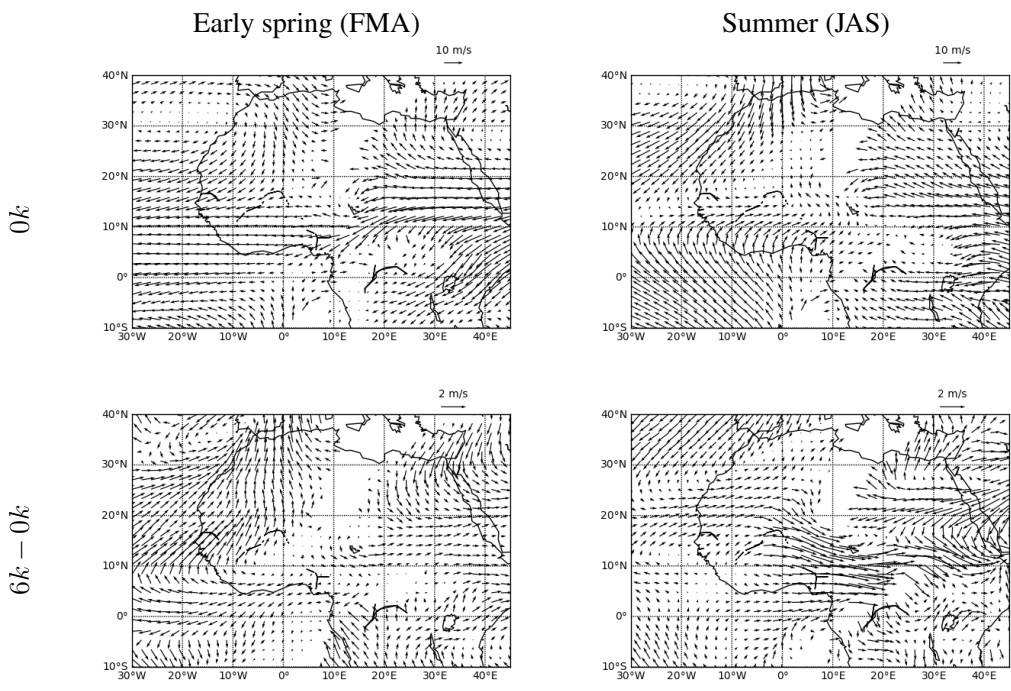

**Figure 11.** Simulated 925 hPa wind speed and directions for early spring (FMA, left) and summer (JAS, right) for 0k and for the difference between 6k and 0k.

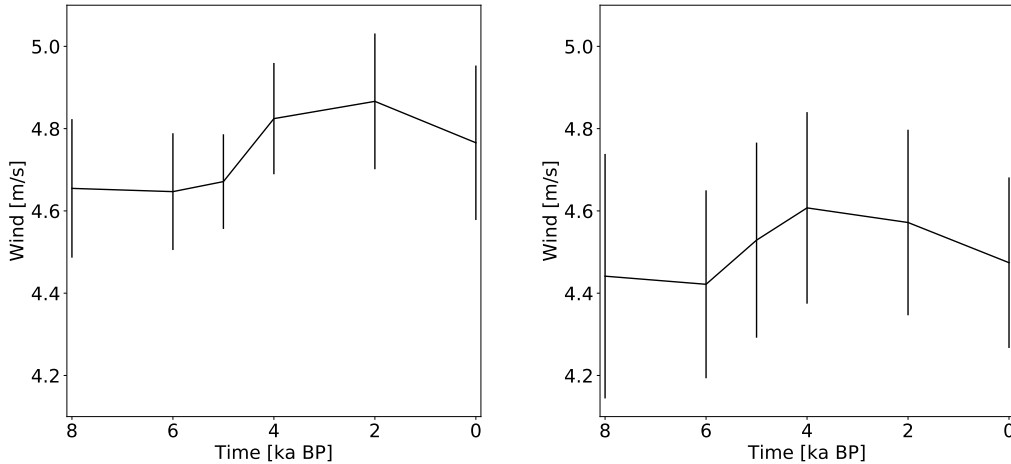

**Figure 12.** Simulated 10m wind strength averaged over the western Sahara (15° W–2° E; 18° N–29° N) for early spring (FMA, left) and summer (JAS, right).

## 4 Discussion and conclusion

We have found a rapid increase in simulated dust deposition along the northwest African margin between 6 and 4 ka BP, which is consistent with the abrupt change in North Atlantic dust deposition records around 20° N at about 5.5 ka BP (deMenocal et al., 2000; Adkins et al., 2006) and 4.9 ka BP (McGee et al., 2013a). The simulated dust deposition agrees quantitatively
well with the data for GC68 and ODP658 at around 20° N (Adkins et al., 2006; McGee et al., 2013a), but is much higher in our simulations than observed for the more northern cores GC37 and GC49 throughout the Holocene. At the remote cores VM20-234 in the tropical North Atlantic, CCG103 close to the Bahamas (Williams et al., 2016) and in the mid-North Atlantic (Middleton et al., 2018), we find a somewhat steeper rise in dust deposition between 6 and 2 ka BP, which is less sharp compared to the rise in dust loading close to the margin. The simulated dust deposition fluxes agree for the mid-Holocene
with the sediment records of Williams et al. (2016) but are higher in the pre-industrial simulation. In the mid-North Atlantic, simulated dust deposition exceeds those indicated in the records (Middleton et al., 2018) by an order of magnitude throughout the Holocene. The discrepancy between simulated dust deposition and data at the more northern cores at the margin and at the more remote cores may result from an overestimation of simulated dust emission. In our model, dried-out paleolakes are prescribed as preferential dust sources, meaning that for these areas the threshold wind velocity, that is necessary to enable
dust emission, is reduced. In contrast, recent satellite observations (Schepanski et al., 2012) indicate that the distribution of main dust sources today is much more diverse. Due to the lower temporal resolution at the northern and the remote cores and due to the long distance between source and sink for the latter, uncertainty is higher and an interpretation for changes in dust deposition remains difficult to assess for those cores.

The particle size distribution in this study is nearly identical for all time slices to those in Egerer et al. (2016) (not shown),
meaning that the discrepancy between simulated and observed main particle size still remains. Due to the smaller particle sizes in the simulations, dust might be transported over longer distances resulting in an overestimation at the more remote cores. As previously noted (Mahowald et al., 2014), the atmospheric surface concentration of dust particles tends to be finer than those deposited at the surface. According to Mahowald et al. (2014), there is a tendency towards a finer distribution of dust at the surface compared to dust deposition in the marine cores. However, this can probably not fully explain the large difference in
particle size and further attention is needed to tackle this issue. In our study, the focus lies on the relative changes in simulated dust deposition fluxes rather than on absolute values, since it remains difficult to compare observations and model results for the same particle size range.

A sudden decline of vegetation cover is key to explain the sudden rise of western Saharan dust emission and the associated rapid increase in North Atlantic dust deposition in our simulations because in a previous study with a prescribed linear decline
of vegetation and lake surface fraction, the simulated dust emission and deposition increased rather linearly (Egerer et al., 2017). Thereby, changes in simulated vegetation cover and dust emission in the western Sahara proceed much more rapidly than the insolation forcing. We find a fast decline of vegetation cover in the western Sahara from 22° N to 18° N in line with a strong reduction of precipitation which points to local vegetation-precipitation feedbacks as suggested by Brovkin et al. (1998) and Claussen et al. (1999). Due to these feedbacks the transition from a wet and vegetated state into a dry desert state may be

accelerated. Still, in our simulations we can not demonstrate vegetation-climate feedbacks to be the definite cause for the rapid decline of vegetation. Alternatively, Liu et al. (2006) proposed a precipitation threshold on vegetation growth to cause a sudden shift in vegetation.

Besides vegetation-climate interactions, changes in lake area cause a rapid dust emission increase in our simulations. The areas of the strongest and fastest change in dust emission are those of former paleolakes. In these areas, lakes disabled dust emission during the mid-Holocene. As soon as the lakes desiccated, fine grained material was favored to be deflated by surface winds, which makes them a highly productive dust source today. However, for the pre-industrial time slice, western Saharan dust emission is likely overestimated in our model due to the prescription of preferential dust sources as discussed above. In contrast, satellite observations show less dust emission from dried-out paleolakes than previously thought (Schepanski et al., 2012). As a consequence, changes in lake surface area play presumably a minor role than our results suggest. In the model, a grid cell where the lake cover fraction exceeds 50% is considered a lake cell and dust emission is not possible. In this way, an artificial threshold on dust emission is created as soon as the lake fraction falls below 50%. This occurs in several grid cells in the western Sahara and the Bodélé Depression, where lake levels were higher than 50% during the mid-Holocene. How does this affect the results of our study? The decline of lake surface area in this study is prescribed in the same way as in our previous study (Egerer et al., 2017), where we found a rather gradual increase of North Atlantic dust deposition despite this artificial threshold. Thus, the sudden decline of vegetation in the western Sahara rather than a gradual desiccation of lakes is likely the trigger of the rapid dust deposition shift that we see in the current simulations. In order to quantitatively separate the effect of a decreasing vegetation cover and the shrinking of lake surface area on the rapid increase of North Atlantic dust deposition, additional sensitivity experiments are necessary.

During the mid-Holocene, the simulated vegetation does not propagate as far north as indicated by pollen records (Fig. 2) and consequently too much dust is emitted from the uncovered soil. Causes for this deficiency might be the under-complex representation of North African vegetation by only a few plant functional types (PFT) in the model (Groner, 2017) or the too simple soil albedo scheme (Vamborg et al., 2011). If the simulated vegetation would shift as far north as indicated by pollen records, we suspect that the decline of vegetation in the western Sahara would be even stronger, resulting in a higher contrast of dust emission and associated North Atlantic dust deposition between the mid-Holocene and the pre-industrial era. Consequently, we expect that the change in western Saharan vegetation, precipitation and dust emission would be at least as abrupt, if not even sharper compared to our current simulations. However, our simulated mid-Holocene vegetation is much more pronounced and shifted northward by about 6° compared to pre-industrial vegetation. Consistent with the mismatch between simulated vegetation and pollen records, ECHAM6-HAM2 is not capable to properly simulate the northward extent of the West African monsoon as indicated by paleo data (Bartlein, 2011). Nevertheless, the mid-Holocene precipitation bias between our simulations and observations is less compared to the results of most other CMIP5 models (Fig. 3). Still, the meridional distribution of precipitation changes is not consistent with observations and requires further attention.

In our simulations, the major changes in western Saharan dust emission occur in early spring (Fig. 10), which can be attributed to changes in dust sources (Fig. 9) but also to a strengthening of the trade winds (Fig. 11). During this season, north-easterly trade winds in the western Sahara and along the northwest African margin are mainly responsible for dust transport to

the North Atlantic (Engelstaedter and Washington, 2007). However, maximum surface concentrations of dust reflecting low-level transport of Saharan air masses are observed in boreal winter (DJF) at Cape Verde Islands (Fomba et al., 2014) and in the eastern tropical North Atlantic (Baker et al., 2013; Powell et al., 2015) for present-day, reflecting uncertainty in the exact timing of maximal dust fluxes. We find that the dynamics of the simulated wind strength averaged over the western Sahara is

similar to the one of simulated dust emission with a rapid and strong shift especially between 5 and 4 ka BP (Fig. 12). The rapid shift in the northeasterly wind strength in this study is consistent with changes in upwelling inferred from SST and biogenic flux records along the NW African margin (Adkins et al., 2006; Romero et al., 2008; Bradtmiller et al., 2016). Furthermore, the surface wind strength is connected to the vegetation via the roughness length. The strong and fast decrease of vegetation in the western Sahara is consistent with the sudden acceleration of surface winds between 5 and 4 ka BP. During the summer,

when dust is transport within the Saharan Air Layer (Prospero et al., 2002; Engelstaedter and Washington, 2007) that is linked to the African Easterly Jet and the West African monsoon system, changes in dust emission in the western Sahara between our simulated time slices are minor. Thus, changes in the atmospheric circulation due to a shift of the summer monsoon are found to be less important in our study compared to the changes of northeasterly winds in the western Sahara during early spring concerning the rapid shift in North Atlantic dust deposition.

In the eastern Sahara, there is only a slight increase in simulated dust emission north of 18° N between 6 and 5 ka BP. The rapid increase in simulated dust deposition in the North Atlantic is hence rather determined by a sudden rise in dust deflation from western Saharan dust sources than by an increase of eastern Saharan dust emission. South of 18° N, there is a strong but gradual decrease of vegetation cover and precipitation roughly between 5 and 2 ka BP. In this area, where trees, grasses and shrubs coexist in our simulations, the higher plant diversity could stabilize the system which results in a more gradual

vegetation decline as shown by modeling studies (Claussen et al., 2013; Groner et al., 2015; Groner, 2017). The vegetation in this area is too dense to enable dust emission until 4 ka BP.

In addition to vegetation-climate feedbacks that might have caused the sudden decrease of vegetation cover in the western Sahara, surface water-climate feedbacks (Krinner et al., 2012), SST-climate feedbacks (Zhao et al., 2005) as well as SST-dust feedbacks (Williams et al., 2016) were proposed as candidates to explain an abrupt end of the AHP. Due to the static

prescription of lakes and SST in our simulations, we could not assess the contribution of these feedbacks to an abrupt change in North Atlantic dust deposition. Based on previous studies (Kutzbach and Liu, 1997; Williams et al., 2016), we expect that the inclusion of an interactive ocean in the model would lead to a further strengthening of the West African monsoon and a northward shift of simulated vegetation during the mid-Holocene. This could help to minimize the gap between simulated vegetation and pollen records, which indicate more extensive precipitation and vegetation north of 20° N compared to our

simulations. Part of the climate-ocean feedback is already taken into account because a change in SST is prescribed based on simulations including an interactive ocean (Bader et al., in prep.). We expect that the transition from the 'green' Sahara to the present-day desert would be even more accelerated by taking all feedbacks into account. Therefore, the pace of the change in simulated North Atlantic dust deposition and Saharan landscape found in this study is presumably still a lower estimate.

While a previous study associated the abrupt shift in North Atlantic dust deposition indicated by sediment records to a large-

scale change in North African landscape and climate conditions (deMenocal et al., 2000), we argue that a spatial distinction of

dust source areas is necessary. Although the sediment cores may be representative for northwestern Africa, they are not capable to explain climate change in the whole area of North Africa. For instance, at specific sites, we see a gradual shift in vegetation cover in our simulations as indicated by pollen data (Kröpelin et al., 2008) and a gradual change in dust emission as seen in reconstructions of aeolian dust accumulation (Cockerton et al., 2014) in contrast to the rapid changes in simulated vegetation

5    cover and simulated dust emission in the western Sahara and the abrupt change in North Atlantic dust deposition. The spatial and temporal heterogeneity in the transition of the North African landscape found in our simulation implies that conclusions from local data records on continental-scale North African landscape and climate change have to be treated with caution.

# 5 Conclusions

To summarize our findings we answer the questions that we posed in the introduction:

- *Can we confirm an abrupt shift in North Atlantic dust deposition in our simulations as found in marine sediment records?*

  We find a rapid shift in North Atlantic dust deposition along the northwest African margin in our simulations about 6 to 4 ka BP in agreement with marine sediment records at around 20° N.

- *How is the shift in North Atlantic dust deposition linked to Saharan landscape and climate transition towards the end of the AHP?*

  The rapid shift in simulated North Atlantic dust deposition during the Holocene is linked to a fast decline of vegetation cover and a strong reduction of lake surface area in the western Sahara accompanied with a fast decline of precipitation and a rapid acceleration of surface winds. The North African drying proceeded thereby much more rapidly than changes in the insolation forcing.

- *How does the timing and the abruptness of Saharan landscape and climate transition vary spatially?*

  Our study emphasizes spatial and temporal heterogeneity in the transition of North African landscape and climate. Implications from local data records on continental-scale North African landscape and climate change have thus to be treated with caution.

*Author contributions.* S.E. and M.C. designed the experiments. S.E. performed and analyzed the simulations and prepared the manuscript. All authors contributed ideas and to the writing of the article and discussed the results.

*Competing interests.* No competing interests are present.

*Acknowledgements.* We thank Anne Dallmeyer (MPI-M) for preparation of the biome maps to evaluate the vegetation distribution. Further we are grateful to Thomas Raddatz and Roberta D'Agostino (both MPI-M) for stimulating discussions. The ECHAM-HAMMOZ model is developed by a consortium composed of ETH Zurich, Max Planck Institut fuer Meteorologie, Forschungszentrum Juelich, University of Oxford, the Finnish Meteorological Institute and the Leibniz Institute for Tropospheric Research, and is managed by the Center for Climate Systems Modeling (C2SM) at ETH Zurich. The service charges for this open-access publication have been covered by the Max Planck Society.

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
