# Peer review of "Rapid increase in simulated North Atlantic dust deposition due to fast change of northwest African landscape during the Holocene"

_Climate of the Past, 2018_

## Referee Comment (RC1) · Anonymous Referee #1 · 27 Apr 2018

Egerer et al. use an atmosphere-aerosol-land model with dynamical vegetation to conduct a series of equilibrium climate simulations for the Holocene, with the aim of analyzing the effects that changes in climate and surface conditions have on dust emissions in North Africa. Model results for vegetation composition and dust are compared to observational data, respectively pollen and sediment records from the NW African continental margin. This work follows two previous studies with similar versions of the same model and different configurations, also dedicated to analyzing variations in the Holocene dust cycle in North Africa. I found the work and the manuscript generally satisfactory, and I only have some minor comments.

[Figure]

2 (19-20). What about Egerer et al. 2017?

3 (3-6). Is dust having any feedback on climate in your simulations?

4 (29-31). You mention initial and boundary conditions. How long have you run your dust experiments from there?

5 (5). Since we cannot see the manuscript in preparation, please spend a couple of words on it.

Figure 5. The model results (absolute values, North-South gradient) are similar to Egerer et al. 2017, but quite different from Egerer et al. 2016. Could you briefly explain what changed? This could be useful in your discussion concerning the apparent over-estimation of deposition fluxes in correspondence of the more northern cores. Also, somewhere in the text please discuss a bit more the data, e.g. what are the assumptions in terms of isolating the dust flux, what size ranges you are comparing, etc.

10 (10-15). Could you calculate dust emission budgets for the two sub-regions and see the absolute and relative changes to support your discussion?

10 (15-20). These explanations all refer to the real world I believe, but what about in your simulations?

10 (19-20). The two potential sources have different composition; but which one does match the chemical composition of dust in those cores you are looking at?

10 (15-20). In general I do not think you really show that one source is dominant for the cores in your model simulations. You may argue explicitly that it seems reasonable to assume so. In this paragraph it seems that model and observations are mixed up, whereas I think you should separate clearly what you can say for each of them, and later discuss if you see convergences and/or differences.

Figure 8. I wonder if you can highlight somehow in the figure the boxes you discuss in the text

19 (13). Please add a concluding paragraph that concisely describes your results with a short summary.

---

## Referee Comment (RC2) · D. McGee (Referee) · 7 May 2018

This paper examines the question of whether abrupt mid-Holocene dust flux increases recorded in NW African margin marine sediment cores reflect similarly abrupt changes in vegetation and precipitation in dust source areas. It also examines which parts of North Africa are represented by these dust flux records. The study centers around time slice experiments spanning 8 ka through the preindustrial that employ dynamic vegetation and interactive dust.

I find this study to be a valuable contribution to our understanding of North African climate changes over the Holocene and of how to interpret dust deposition records

in this region. There are a few areas where I think the study could be made more complete and where the writing could be clearer, which I list below, but overall I find this to be worthy of publication after revision.

1. What does modeled dust deposition look like downwind in the Bahamas or in the central tropical/subtropical Atlantic? It would be useful to compare against the records of Williams et al. 2016 and Middleton et al. 2018 as well, especially given that dust deposition in the Bahamas should reflect summer deposition rather than winter/spring deposition, and all the distal sites presumably reflect a broader range of sites than the NW African margin sites.

2. Section 2.2/Section 3.1: For the comparison with coretop dust fluxes, what grain sizes are being transported in the model? Grains at these sites are quite coarse (10-40 $\mu$m modal grain sizes for sediment inferred to be windblown dust), so if the model is getting the right answer for the right reason at ODP658 and GC68, it should be depositing quite coarse dust at these sites. Albani et al. simulated only <10 $\mu$m grains, so it was necessary to compare against fluxes of <10 $\mu$m dust at these sites rather than the total dust fluxes reported by McGee et al. Related to this point, I'm surprised that the coastal dune fields in Mauritania/Western Sahara (e.g., Lancaster et al., Geology 2002) aren't dust sources in the model, as I've always thought that the coarse grains deposited at these sites must have a quite local source.

3. There should be a more detailed and complete summary of modern observations of the seasonality and flux of dust in the region – for example, see R.F. Anderson et al., Phil. Trans. A 2016, especially citations 33-35 for studies of dust deposition in the eastern tropical North Atlantic and at Cape Verde. The work of Skonieczny et al. is also useful for documenting modern transport from dust sources in the NW Sahara.

4. The discussion of wind changes should be expanded and clarified (admittedly, winds are a bit of a fixation for me.) First, winds are only shown for the two timeslices (Figure 10) rather than plotted as a timeseries as is done for the dust, vegetation and precipitation. It would be fairly easy to plot January-through-April northeasterly wind strength over the some portion of the NW Sahara to see whether similarly abrupt changes occur in winds between 6-4 ka. Second, it should be noted in the text that the large changes that the authors find in FMA northeasterly winds are consistent with the changes in upwelling inferred from SST and biogenic flux records along the NW African margin (Adkins et al., 2006; Bradtmiller et al., 2016; Romero et al., 2008) (note that these changes in upwelling proxies are as abrupt as the changes in dust fluxes at these cores). Third, on page 18 lines 14-20, the authors first state that the FMA wind changes are potentially as important as the vegetation and precipitation changes, then end the paragraph by saying that "changes in atmospheric circulation due to a shift of the monsoon system are of minor importance concerning the rapid shift in North Atlantic dust deposition." Aren't the winter/spring wind changes a part of the monsoon system? Or does this second statement just focus on the summer monsoon? If so, that should be clearly stated.

5. I agree with the final point of the paper, that dust fluxes at NW African margin sites reflect conditions in specific NW African dust source areas, and they are not representative of the whole of North Africa. That said, Figure 7 suggests that the dust changes are at least representative of an area 17ËŽ (east-west) by 11ËŽ (north-south) – quite a large part of NW Africa. So I think this statement should be qualified – the dust records are representative of a large area, just not as large as has sometimes been implied or stated.

6. I think the paper's other main point is that, at least in this model, the rapid increases in dust deposition recorded in these sites require rapid decreases in precipitation and vegetation density (and perhaps rapid increases in winter/spring winds) in NW Africa – the rapid dust changes cannot be attributed to thresholds inherent to dust emission superimposed on gradual changes in climate and vegetation. The study thus suggests that mid-Holocene drying of NW Africa proceeded much more rapidly than the decline in insolation would suggest, correct? If I have this correct, I think this second point

could be stated more clearly and emphasized in the text.

Other comments:

Page/line:

P1/L6: Here and in page 5/line 4, "Therefore" is used improperly. It should be used to mean "Because of this", but in both places it is used in place of "In order to do this". Please change to "To do this" or equivalent.

P8/L5: "At the more northern cores GC37 and GC49 the change in dust deposition is rather moderate." If this statement is intended to mean that the magnitude of the dust deposition change is smaller at GC37 and GC49, I disagree: see McGee et al. 2013 Figure 5, which shows that relative increases in dust fluxes are similar at GC37, 49, and 68 (the relative change is smaller at ODP658, presumably because this record is bulk terrigenous flux and so overestimates fluxes during the AHP.) If this statement is intended to mean that the rate of change is slower at GC37 and GC49, I also disagree: McGee et al. 2013 demonstrates that the smoother changes recorded at GC37 and GC49 could just be due to bioturbation (greater smoothing of the record due to lower sedimentation rates), not a slower rate of change in dust deposition.

-David McGee

---

## Editor Comment (EC1) · D.-D. Rousseau (Editor) · 7 May 2018

Dear authors,

Both reviewers have released their comments and reviews. As the discussion phase is still opened, I would encourage you to post early comments about these two reviews so that one could use the opportunity that the discussion phase allows the author to exchange with the reviewers before the final decision. All the very best,

denis-didier Rousseau, Clim Past co-editor in chief

---

## Author Comment (AC1) · 25 May 2018

We thank the reviewer for constructive comments and helpful suggestions.

2 (19-20). What about Egerer et al. 2017?

It makes sense to include Egerer et al. 2017 if we talk about the Holocene instead of the mid-Holocene period since Egerer et al. 2017 focus on the dynamics of the whole Holocene period. We will change the sentence accordingly.

3 (3-6). Is dust having any feedback on climate in your simulations?

Yes, indeed the impact of dust on the solar radiation budget through scattering and

absorption and the alteration of the cloud structure is implement in the model (for details see Stier et al. 2005). We will highlight this explicitly in the model description. For the analysis, the feedback of dust on climate can nevertheless not be quantified. To do so, additional sensitivity experiments would be necessary (which we already mentioned in the discussion).

4 (29-31). You mention initial and boundary conditions. How long have you run your dust experiments from there?

If we understood the question correctly, the answer is given in 4 (15-17): To achieve an equilibrium for the vegetation distribution for each time slice, we first run 30 years with an accelerated vegetation dynamics followed by 570 years with vegetation dynamics at normal speed. We take the last 200 years for evaluation. To avoid confusion, we will shift this paragraph to the above mentioned passage.

5 (5). Since we cannot see the manuscript in preparation, please spend a couple of words on it.

Recently, the manuscript was published in Climate of the past and all details can be found in there. We will add the citation (Dallmeyer, A., Claussen, M., and Brovkin, V.: Harmonizing plant functional type distributions for evaluating Earth System Models, Clim. Past Discuss., https://doi.org/10.5194/cp-2018-41, in review, 2018).

Figure 5. The model results (absolute values, North-South gradient) are similar to Egerer et al. 2017, but quite different from Egerer et al. 2016. Could you briefly explain what changed? This could be useful in your discussion concerning the apparent overestimation of deposition fluxes in correspondence of the more northern cores.

The progressive development and installation of new hardware and software at the Max Planck Institute for Meteorology demanded the use of different model versions for the different studies. Older model versions were used in Egerer et al. (2016) (echam6.1-ham2.1) and Egerer et al. (2017) (echam6.1-ham2.2). In the new model

version (echam6.3-ham2.3), the definition of the roughness length is no longer depending on the orography, but on the leaf area index of the vegetation. Due to this changes, a regional modeling factor was introduced in the new model version. So on the one hand the differences arise from different tuning of the dust module. On the other hand, note that the prescription of the vegetation distribution is quite different in Egerer et al. (2016) and Egerer et al. (2017). And both differ from the vegetation distribution which we get in this study with dynamic vegetation. Since dust emission is highly depending on the vegetation distribution, we think that in terms of absolute values it is inappropriate to compare the three studies.

Also, somewhere in the text please discuss a bit more the data, e.g. what are the assumptions in terms of isolating the dust flux, what size ranges you are comparing, etc.

The dust flux was calculated as the difference between the total flux and the carbonate, opal and organic carbon flux. In McGee et al. (2013) and Adkins et al. (2006), the 230Th normalization method was used to determine dust fluxes. Also, McGee et al. (2013) use endmember modelling to separate eolian and hemipelagic fluxes. Thereby, the coarse endmembers (approximately between 8 $\mu$m and 30 $\mu$m in size) are assumed to characterize eolian dust. We will add this information to the 'Comparison with marine sediments' section. Model description: The size distribution of the emitted particles is prescribed via log-normal functions of a coarse (mass mean radius (mmr) = 1.75 $\mu$m, standard derivation $\sigma$ = 2 $\mu$m) and an accumulation mode (mass mean radius (mmr) = 0.37 $\mu$m, standard derivation $\sigma$ = 1.59 $\mu$m). We will add mmr and $\sigma$ in the model description. The problem of differing size ratios between simulated and observed fluxes was already discussed in Egerer et al. (2016) and in earlier studies (e.g. Mahowald et al. (2014). Thus, we will refer to this studies in the manuscript.

10 (10-15). Could you calculate dust emission budgets for the two sub-regions and see the absolute and relative changes to support your discussion?

In Fig. 1, the dust emission budgets for the western Sahara (as in Fig. 7 in the manuscript) and for the area around the Bodele Depression (12°N-18°N, 14°E-21°E) are plotted. This supports the argument, that emissions from the Bodele Depression are rarely significant for changes between 6 and 4 ka BP and occur mainly after 4 ka BP. We will add Fig. 1 to the manuscript and thank the reviewer for the suggestion.

10 (15-20). These explanations all refer to the real world I believe, but what about in your simulations? 10 (15-20). In general I do not think you really show that one source is dominant for the cores in your model simulations. You may argue explicitly that it seems reasonable to assume so. In this paragraph it seems that model and observations are mixed up, whereas I think you should separate clearly what you can say for each of them, and later discuss if you see convergences and/or differences.

In the simulations, it is unfortunately not possible to track dust particles from source to sink since the tracers for dust are interacting with surrounding particles and can not be extracted. To support the argumentation, we will include Fig. 1 to the section. We think that the discussion (also involving 'real world believes') is necessary to be conducted in this section because the choice of the dust emission area that is responsible for dust deposition at the margin and the next chapter is based on that discussion. We will nevertheless separate the arguments arising from our simulations and previous observations more clearly.

10 (19-20). The two potential sources have different composition; but which one does match the chemical composition of dust in those cores you are looking at?

Unfortunately, in the paper which we referred to, this information is not given. We were pointing to the approach that if the chemical composition is significantly different in the two source areas, it must be clearly differentiable in the sink area. Since this is a vague argument, it might be better to remove this sentence from the manuscript.

Figure 8. I wonder if you can highlight somehow in the figure the boxes you discuss in the text.

Yes, this seems like a good idea. We will mark the boxes of strong changes in dust emission and whether they coincide with a strong change in lake area, vegetation cover or both with different colors.

19 (13). Please add a concluding paragraph that concisely describes your results with a short summary.

To summarize our findings we will give a short conclusion where we answer the questions that we posed in the introduction.
* * *
[Figure]

[Figure]

**Fig. 1.** Simulated dust emission from the western Sahara and from an area including the Bodele Depression.

[Figure]

---

## Author Comment (AC2) · 25 May 2018

We thank David McGee for his constructive comments and helpful suggestions.

1. What does modeled dust deposition look like downwind in the Bahamas or in the central tropical/ subtropical Atlantic? It would be useful to compare against the records of Williams et al. 2016 and Middleton et al. 2018 as well, especially given that dust deposition in the Bahamas should reflect summer deposition rather than winter/spring deposition, and all the distal sites presumably reflect a broader range of sites than the NW African margin sites.

[Figure]

In my doctoral thesis (Egerer, S. (2018). Linking marine dust records to Saharan landscape evolution during the Holocene: a theoretical study. Phd Thesis, Hamburg: Universität Hamburg. doi:10.17617/2.2552057), I additionally compared to the Bahama and central/tropical Atlantic core sites. However, in this study, we decided to focus on the dust cores close to the margin. The temporal resolution of the core sites further away from the margin is much coarser and thus uncertainty is much higher. Also, we would like to focus more on the abrupt change at the core sites close to the margin and would thus like to restrict this study to these sites.

2. Section 2.2/Section 3.1: For the comparison with coretop dust fluxes, what grain sizes are being transported in the model? Grains at these sites are quite coarse (10-40 $\mu$m modal grain sizes for sediment inferred to be windblown dust), so if the model is getting the right answer for the right reason at ODP658 and GC68, it should be depositing quite coarse dust at these sites. Albani et al. simulated only <10 $\mu$m grains, so it was necessary to compare against fluxes of dust at these sites rather than the total dust fluxes reported by McGee et al. Related to this point, I'm surprised that the coastal dune fields in Mauritania/Western Sahara (e.g., Lancaster et al., Geology 2002) aren't dust sources in the model, as I've always thought that the coarse grains deposited at these sites must have a quite local source.

The grain size distribution in this study is quite similar to Egerer et al. (2016), which peaks around 6-7 $\mu$m. Already in Egerer et al. (2016), we discussed possible reasons and consequences of the discrepancy between the grain sizes in the simulation and observed particle sizes. Despite the mismatch, we think that it is valid to compare both, because we think the fluxes <10 $\mu$m are proportional to the total amount of deposited dust. We are afraid our global circulation model with a grid cell length of about 200km is not suitable to resolve and reflect dune field activity. As was discussed in our manuscript, the dust source areas in the model are in conflict with satellite observations (Schepanski et al., 2009) and thus might not reflect dust source areas realistically.

3. There should be a more detailed and complete summary of modern observations

of the seasonality and flux of dust in the region – for example, see R.F. Anderson et al., Phil. Trans. A 2016, especially citations 33-35 for studies of dust deposition in the eastern tropical North Atlantic and at Cape Verde. The work of Skonieczny et al. Is also useful for documenting modern transport from dust sources in the NW Sahara.

We will involve these studies in the chapter 'Changes in the seasonal cycle of dust emission and atmospheric circulation'. Here it is mentioned, that maximum surface dust concentrations at the Cape Verde Islands are found in boreal winter (DJF), whereas in our simulations dust emissions peak later in early spring (FMA).

4. The discussion of wind changes should be expanded and clarified (admittedly, winds are a bit of a fixation for me.) First, winds are only shown for the two timeslices (Figure 10) rather than plotted as a timeseries as is done for the dust, vegetation and precipitation. It would be fairly easy to plot January-through-April northeasterly wind strength over the some portion of the NW Sahara to see whether similarly abrupt changes occur in winds between 6-4 ka. Second, it should be noted in the text that the large changes that the authors find in FMA northeasterly winds are consistent with the changes in upwelling inferred from SST and biogenic flux records along the NW African margin (Adkins et al., 2006; Bradtmiller et al., 2016; Romero et al., 2008) (note that these changes in upwelling proxies are as abrupt as the changes in dust fluxes at these cores). Third, on page 18 lines 14-20, the authors first state that the FMA wind changes are potentially as important as the vegetation and precipitation changes, then end the paragraph by saying that "changes in atmospheric circulation due to a shift of the monsoon system are of minor importance concerning the rapid shift in North Atlantic dust deposition." Aren't the winter/spring wind changes a part of the monsoon system? Or does this second statement just focus on the summer monsoon? If so, that should be clearly stated.

We will add a time series of the 10m wind strength over NW Africa during early spring (FMA) (Fig. 1). (Note: The shape is in principal equal to Jan-April but we chose FMA for consistency with Fig.11). We thank you for this suggestion. Indeed, there is a clear

rapid rise of the wind strength between 6 and 4 ka BP in the western Sahara in line with the increase in dust emission. We will clearly state the relation between changes in wind strength and dust emission in the text. Furthermore, the wind strength is linked to vegetation through the roughness length. The decrease in vegetation cover is in line with an acceleration of surface winds. We will also underline the consistency between the FMA northeasterly winds and changes in upwelling inferred from SST and biogenic flux records not only in strength but also in speed. Yes, indeed we refer here to the summer monsoon and will change this sentence accordingly.

5. I agree with the final point of the paper, that dust fluxes at NW African margin sites reflect conditions in specific NW African dust source areas, and they are not representative of the whole of North Africa. That said, Figure 7 suggests that the dust changes are at least representative of an area 17°(east-west) by 11° (north-south) – quite a large part of NW Africa. So I think this statement should be qualified – the dust records are representative of a large area, just not as large as has sometimes been implied or stated.

We will specify the statement accordingly in the discussion.

6. I think the paper's other main point is that, at least in this model, the rapid increases in dust deposition recorded in these sites require rapid decreases in precipitation and vegetation density (and perhaps rapid increases in winter/spring winds) in NW Africa – the rapid dust changes cannot be attributed to thresholds inherent to dust emission superimposed on gradual changes in climate and vegetation. The study thus suggests that mid-Holocene drying of NW Africa proceeded much more rapidly than the decline in insolation would suggest, correct? If I have this correct, I think this second point could be stated more clearly and emphasized in the text.

Your argument is correct and we will emphasize the stated point more clearly in the discussion.

Other comments:

Page/line: P1/L6: Here and in page 5/line 4, "Therefore" is used improperly. It should be used to mean "Because of this", but in both places it is used in place of "In order to do this". Please change to "To do this" or equivalent.

We agree and will change the phrases accordingly.

P8/L5: "At the more northern cores GC37 and GC49 the change in dust deposition is rather moderate." If this statement is intended to mean that the magnitude of the dust deposition change is smaller at GC37 and GC49, I disagree: see McGee et al. 2013 Figure 5, which shows that relative increases in dust fluxes are similar at GC37, 49, and 68 (the relative change is smaller at ODP658, presumably because this record is bulk terrigenous flux and so overestimates fluxes during the AHP.) If this statement is intended to mean that the rate of change is slower at GC37 and GC49, I also disagree: McGee et al. 2013 demonstrates that the smoother changes recorded at GC37 and GC49 could just be due to bioturbation (greater smoothing of the record due to lower sedimentation rates), not a slower rate of change in dust deposition.

We intended to say that changes in dust flux are not as sharp as for the more southern core sites. From McGee et al. it is not clear, whether there is indeed a slower rate of change in dust deposition or if a smoothing due to bioturbation is indeed the cause of the slower change rate.

―――――――――――――――――――

[Figure]

**Fig. 1.** Time series of 10m wind strength over NW Africa during early spring (FMA).

---

## Author Response (AR2)

Response to Referee #1:

*The response of Egerer et al. to my previous comments are generally satisfactory and address the questions that were raised in the review process. I only have one remark at this point.*
*At page 19 lines 12-15 the authors report, attributing the statement to Mahowald et al. (2014), that "dust particles deposited at the surface are in general finer than those deposited in marine cores". I think the authors here erroneously reported what should be something like "dust particles in the atmospheric surface layer are finer than those deposited to the surface", i.e. as an effect on preferential gravitational settling of larger particles – please correct this.*

- ☐ We agree that Mahowald et al. (2014) state that "dust particles in the atmospheric surface layer are finer than those deposited to the surface". But they also conclude therefrom that there is a tendency towards a finer distribution of dust at the surface compared to dust deposition in the marine cores. We will state this argument more precisely in the manuscript.

*Linking this to a comment by David McGee on the size range, I also find that the implication is that relative changes is the simulated dust deposition fluxes are a more robust result of this work, compared to the absolute values of dust deposition fluxes, since a coherent comparison of observations and model results on the same particle size range is not carried out.*

- ☐ Yes, indeed this is the case and we state this explicitly in the discussion:
- ☐ In our study, the focus lies on the relative changes in simulated dust deposition fluxes rather than on absolute values, since it remains difficult to compare observations and model results on the same particle size range.

Response to Referee #2:

*I appreciate Egerer et al.'s consideration of my comments, and in particular the addition of the wind-related figures and discussion. The only comment that I do not feel is adequately addressed is my request that the model results be compared to the three records available from the central and western North Atlantic – the two in Williams et al. 2016 and the record from Middleton et al. EPSL 2018. In particular, the Williams et al. record from the Bahamas is similar in temporal resolution to the records from the African margin and also shows a steep rise in dust fluxes beginning around 5 ka, so it seems highly related to the manuscript. It seems important to know whether the model a) produces dust deposition changes similar to those observed in distal North Atlantic sites and b) predicts a similarly abrupt rise in dust loading over most of the subtropical Atlantic, or whether the abrupt rise is confined to the African margin. If the model doesn't reproduce the observed changes at distal sites, this seems important to note as an area for future investigation. I do not think it is sufficient to say that the results are in an unpublished thesis.*

- We added the comparison to the Williams and Middleton dust cores and expanded the discussion accordingly.
- For your interest: The thesis is published already and available on the webpage of the Max Planck Institute for Meteorology:

https://www.mpimet.mpg.de/en/science/publications/reports-on-earth-system-science/

*Finally, one area for clarification: in Section 3.1, the authors should be clear about whether the dust fluxes used for GC37, GC49 and GC68 are the total eolian fluxes from McGee et al. 2013, or the <10 μm eolian fluxes given in Albani et al. 2015.*

*-David McGee*

- The dust fluxes we used are the total eolian fluxes. We state this in Section 3.1 with a short comment about the Albani et al. (2015) dataset.
- See also the short comment which we gave to  Referee #1 on the comparison of observations and model results concerning the particle size distribution and which we included in the discussion section:

[revised manuscript text omitted]